# The Role of the OLM CandID Real-Time PCR in the Invasive Candidiasis Diagnostic Surveillance in Intensive Care Unit Patients

**DOI:** 10.3390/microorganisms13030674

**Published:** 2025-03-18

**Authors:** Laura Trovato, Maddalena Calvo, Concetta Ilenia Palermo, Maria Rita Valenti, Guido Scalia

**Affiliations:** 1Department of Biomedical and Biotechnological Sciences, University of Catania, 95123 Catania, Italy; maddalenacalvo@gmail.com (M.C.); lido@unict.it (G.S.); 2U.O.C. Laboratory Analysis Unit, A.O.U. Policlinico “G. Rodolico-San Marco” Catania, 95123 Catania, Italy; ci.palermo@policlinico.unict.it; 3Department of Anesthesiology and Intensive Care, A.O.U. Policlinico “G. Rodolico-San Marco” Catania, 95123 Catania, Italy; mritavalenti@hotmail.it

**Keywords:** *Candida* spp., invasive candidiasis, molecular diagnosis, serum real-time PCR, candidaemia, BDG

## Abstract

Molecular techniques recently integrated the candidiasis diagnostic workflow, avoiding the culture-based prolonged turn-around time and lack of sensitivity. The present retrospective study evaluated the OLM CandID Real-Time PCR on serum samples in the early and rapid candidaemia diagnosis among ICU patients. The final purpose of the protocol was to demonstrate the effectiveness of a PCR assay in the invasive candidiasis diagnostic workflow due to the high sensitivity rates and species identification possibility. The evaluation screened 60 suitable patients, accounting for 10 probable and 7 proven candidiasis cases. Patients with at least a positive (1→3)-β-D-glucan (BDG) value underwent molecular procedures. A sensitivity of 83.3%, a specificity of 94.3%, a positive predictive value of 87.5%, and a negative predictive value of 91.7% emerged for the PCR assay. As a conclusion, *Candida* PCR assays may represent useful diagnostic assistance tools when applied together with serological markers and culture-based assays.

## 1. Introduction

*Candida* species colonises the respiratory, genital, and cutaneous tracts within the human host, whose specific conditions may enable fungal dissemination. Invasive candidiasis (IC) episodes represent a concerning healthcare challenge due to a broad symptom spectrum and the variable diffusion of *Candida* species into different geographical areas [1,2,3]. Epidemiological studies register 25,000 candidaemia cases each year [2]. The candidaemia mortality rate statistically overcomes a 70% value, specifically involving intensive care unit (ICU) patients and immunocompromised hosts [1]. Worldwide data document significant healthcare cost (up to USD 29,000) and hospitalisation length (up to 13 additional days) increases due to invasive candidiasis [2]. The prevention of the dissemination and investigations about *Candida* spp. may become essential in managing invasive candidiasis cases. On the other hand, a prompt diagnosis is a fundamental action to guarantee effective patient management, thereby reducing the morbidity and mortality percentages [3].

*Candida* spp. bloodstream invasion is a concrete possibility in all high-risk cases, such as multi-site *Candida*-colonised cases, severe neutropenic patients, previous surgical patients, and patients treated with broad-spectrum antibiotics [4]. Culture methods still represent the gold standard in fungal infection diagnosis [5].

Unfortunately, these techniques suffer from low sensitivity (50%), often delaying the diagnostic workflow and the antimycotic application. For instance, a blood culture exam gathers a sensitivity equal to one colony-forming unit (CFU)/mL, even requiring several days (5) to furnish a definitive result [6]. Although culture results are fundamental to prove candidiasis, a presumptive invasive fungal infection diagnosis may include serological biomarkers and clinical evidence. As regards this purpose, several clinical studies confirm the usefulness of the (1→3)-β-D-glucan (BDG) assay among ICU patients. This assay may express a significant variability in terms of the specificity and reproducibility, depending on the assessed reagent kit and the involved patient [7].

Previous studies have demonstrated how high BDG levels (>259 pg/mL) allow for the discrimination between *Candida* spp. colonisation and invasive candidiasis. A positive immunofluorescent *Candida albicans* germ tube title (1:160) may support this diagnosis in critical patients [8]. The fungal DNA detection recently integrated the diagnostic workflow, avoiding incubation or culture phases. Most protocols include multiple targets based on the 18S rDNA, the 28S rDNA, or the mitochondrial DNA. Serum, blood, and plasma samples are possible starting matrices for these methods [5]. Despite the variability in precocity and accuracy, these samples are suitable to ensure early candidaemia detection [5]. For instance, magnetic resonance based its detection capability on blood samples, reaching high sensitivity rates during reduced time intervals (3–5 h) [9]. Additionally, Real-Time PCR has been extensively experimented with in detecting *Candida* species both for serum and plasma samples, documenting high accuracy and agreement rates with the conventional diagnostic procedures [10]. The present study evaluates the role of the OLM CandID Real-Time PCR (LionDx srl, Pordenone, Italy) in the early and accurate candidaemia diagnosis among ICU patients.

## 2. Materials and Methods

### 2.1. General Characteristics of the Study

A retrospective observational study was conducted at the University Hospital Policlinico (Gaspare Rodolico) of Catania over eleven months (July 2023–June 2024) in the Intensive Care Unit setting. The study included all of the patients with prolonged ICU stay (>4 days) and the presence of one or more risk factors, such as broad-spectrum antibiotic therapy, haematological diseases, prolonged corticosteroid usage, and a *Candida* colonisation index (CI) value ≥ 0.5. Patients with ICU stays equal to or shorter than 4 days were excluded from the study. Additionally, patients with prolonged ICU stay (>4 days) in the absence of at least one of the above-mentioned risk factors were not included.

### 2.2. Retrospective Data Collection

The study did not require supplementary biological samples because ICU routine surveillance programs include weekly serological markers detection and consequent serum storage at −80 °C. Serum samples were restored and processed through the experimental molecular assay only for the eligible patients. These patients demonstrated at least a positive BDG value after the Fungitell Assay (Associates of Cape Cod) according to the manufacturer’s instructions. Surveillance cultures for *Candida* spp. colonisation were performed once a week.

These cultures required different samples, such as bronchial aspirates, and urine and rectal swabs, which underwent a 48–72 h incubation at 37 °C on Sabouraud Dextrose agar (Vakutest Kima, Arzergrande, Italy). Certainly, surveillance cultures result only in the suggested *Candida* spp. colonisation, warning clinicians about possible fungal dissemination (endogenous infections).

The colonisation index was calculated as the ratio of culture-positive surveillance sites to the total number of screened sites [10]. Blood cultures were obtained at the discretion of the attending physician (systemic infection symptoms and positive inflammatory indices) and were processed using the automated BACTEC system (Becton Dickinson, Franklin Lakes, NJ, USA). Specifically, all of the proven- and probable-candidiasis patients underwent a blood culture collection due to their signs, symptoms, colonisation indices, and risk factors. Blood culture samples were collected at the same time of the serum sample to perform serological analysis among these specific patients.

### 2.3. Molecular Test Protocol and Patients’ Classification

The serum samples underwent molecular assay through the OLM CandID Real-Time PCR (LionDx srl, Pordenone, Italy). After an automated extraction protocol through the NUCLISENS^®^ EASYMAG (Biomerieux, Florence, Italy), the amplification was conducted according to the LionDx manufacturer’s instructions. The kit includes two different primer mixes (CandID and CandID Plus) together with specific probes for the *C. albicans*, *C. tropicalis*, *C. glabrata* (currently also known as *Nakaseomyces glabratus*), *C. krusei* (currently known also as *Pichia kudriavzevii*), *C. parapsilosis*, and *C. dubliniensis* DNA detection within approximately 40 PCR cycles. Overall, the results emerged within approximately one hour.

All of the included patients were classified as proven or probable-candidiasis cases according to the European Organisation for Research and Treatment of Cancer and the Mycoses Study Group Education and Research Consortium (EORTC) [11], whose criteria were also applied by Bassetti et al. within ICU settings [12]. Remarkably, proven candidiasis requires a positive *Candida* spp. blood culture result.

Otherwise, probable candidiasis was defined by mycological evidence (at least two consecutive positive BDG detections or positive magnetic resonance *Candida* results), host factors (haematological diseases, prolonged corticosteroids usage, *Candida* colonisation, or documented organ transplants), and clinical features (organ abscesses or lesions).

The authors collected all of the BDG values during the candidiasis patients’ hospital stay period within a database. Therefore, we registered the highest BDG value and the average BDG concentration. The highest BDG value represented the suitable serum sample for the PCR assay. The ICU stay length (days), underlying condition, blood culture results, and molecular assay results were documented for the candidiasis cases. As regards the risk factors, sex, age, ICU stay length, previous surgery, antimicrobial treatments, and steroid usage were documented for proven candidiasis, probable candidiasis, and no-candidiasis cases. Specifically, the authors reported information about the patients’ sex due to previous literature data documenting male sex as an invasive candidiasis risk factor [13].

### 2.4. Statistical Analysis

The statistical significance of all of these risk factors was calculated depending on the patient’s classification. The statistical association of BDG, molecular results, and colonisation indices to the candidiasis or non-candidiasis patients was reported. The analysis was performed using the MedCalc Statistical Software version 17.9.2 (MedCalc Software bvba, Ostend, Belgium; http://www.medcalc.org; 2017) and reporting the corresponding *p* values. Specifically, the χ^2^ and Fisher’s exact test were applied to establish the categorical variables as percentages. Medians with ranges were used to describe non-normally distributed continuous variables and compared using the Mann–Whitney U-test. The same software allowed for the calculation of the sensitivity (SS), specificity (SP), positive predictive value (PPV), and negative predictive value (NPV) percentages for a single BDG, duplicate BDG, and PCR assay.

### 2.5. Ethical Aspects

Globally, the study did not directly involve human beings and included actions only on biological samples. All of the information about age, risk factors, sex, colonisation data, antimicrobial treatments, and ICU stay length anonymously emerged from the laboratory informatic system (LIS) without any medical record consultation. Consequently, ethical approval was not mandatory according to the local legislation.

## 3. Results

A total number of 60 critically ill patients were included in the study and a total of 235 serum samples (mean 3.9 per patient, range 1–25) were collected for the BDG detection assay. A total of 109 serum samples revealed a positive BDG value and, of these, 53 samples were selected for the PCR assay. The characteristics and risk factors of the patients are shown in Table 1. In particular, broad-spectrum antibiotic treatment (*p* < 0.01), surgery (*p* < 0.05), and antifungal treatment (*p* < 0.001) were significantly higher in the group of proven/probable IC than the non-candidiasis one.

According to the patients’ classification through the EORTC criteria, 17 (28.3%) patients belonged to the proven and probable invasive candidiasis classification. Clinical and microbiological details about these patients are summarised in Table 2. Specifically, *C. albicans* emerged as the most isolated species (47.0%), while only one case (5.88%) reported a *C. tropicalis* aetiology. The prevalence of documented proven invasive candidiasis was 11.7%, accounting for seven candidaemia episodes (three with *Candida albicans*, two with *Candida tropicalis*, and two with *Candida parapsilosis*). All of these patients had positive BG and OLM CandID Real-Time PCR assay results.

Specifically, patients with proven candidiasis showed an average BDG concentration equal to 442 pg/mL (with a range of 80–920 pg/mL). Patients reporting *Candida* spp. positive blood cultures were also positive after the OLM CandID Real-Time PCR assay, demonstrating a 100% concordance between the culture identification and molecular species detection. Ten patients were classified with probable IC. Particularly, five patients with respiratory failure had fever (a body temperature higher than 38 °C), which was persistent despite the use of broad-spectrum antibiotic treatment for 96 h. One patient with chronic renal failure suffered from the same condition. These six patients’ surveillance cultures were positive for the same *Candida* species from at least two non-contiguous anatomical sites. Three patients with clinical signs of septic shock showed a body temperature higher than 38 °C, a prolonged (more than 7 days of the preceding 30 days) glucocorticoid therapy, the same *Candida* spp. isolation from at least two non-contiguous sites, and a positive BDG value from two consecutive serum samples. Finally, one patient had a haematological malignancy, documenting small abscesses in the liver and a positive BDG value from two consecutive serum samples. All of the patients with probable candidiasis had a positive BG with an average BDG concentration equal to 196 pg/mL (range, 80–323).

A total of 7 patients (70%) reported a positive OLM CandID Real-Time PCR assay result among the total number (10) of probable-candidiasis cases. Figure 1 indicates the distribution of the BDG concentration among no-candidiasis and proven/probable-candidiasis patients.

The distribution suggests how BDG concentrations are significantly higher in proven/probable-candidiasis patients (median 299 pg/mL) than in no-candidiasis cases (median 119 pg/mL). Consequently, we attributed a statistical significance (*p* < 0.05) to the highest BDG concentrations, which were related to the candidiasis condition.

Patients with proven/probable IC were compared to patients who had no identified IC according to the modified EORTC/MSG criteria. Twenty-four patients (46.1%) had PCR-positive results. Among the no-candidiasis cases, 35 patients showed BDG-positive results, while only 2 (3.8%) of the 52 tested patients reported a PCR-positive result. The single positive BDG result had a sensitivity of 100%, a specificity of 18.6%, a PPV of 32.7%, and an NPV of 100%. Otherwise, the duplicate positive BDG result had a sensitivity of 88.2%, a specificity of 76.7%, a PPV of 60%, and a NPV of 94.3%. Finally, the PCR assay for the diagnosis of the proven/probable IC documented a sensitivity of 83.3%, a specificity of 94.3%, a PPV of 87.5%, and an NPV of 91.7%. Duplicate positive BDG results (*p* < 0.0001) and positive PCR assays (*p* < 0.0001) were significantly higher in the proven/probable IC patients’ group. As a result, all of the proven candidiasis cases (100%) with a positive blood culture matched the positive PCR assay result. As regards the probable candidiasis, a negative blood culture result appeared for seven patients (70%), together with a negative molecular *Candida* spp. reporting. On the other hand, three probable-candidiasis cases (30%) appeared negative after the blood culture and the molecular methods. Table 3 reports a comparison between the positive PCR assay cases and the blood culture results, along with the BDG values.

## 4. Discussion

We proposed this retrospective study to demonstrate the relevant value of Real-Time PCR in the early invasive candidiasis diagnosis. The study showed high sensitivity rates and highlighted the significant agreement between the molecular techniques and conventional diagnostic methods. Invasive candidiasis holds primacy in clinical adverse outcomes due to diagnostic delays and potential microbial virulence. Early species identification reduces the mortality, especially in intensive care patients, whose risk factors complicate systemic infection development and management [10,11,12,13,14,15]. According to the scientific evidence, it is essential to deeply analyse all of the possible patient risk factors, defining which patients are more susceptible to *Candida* spp. infection. Consequently, the reporting of antimicrobial treatment, steroid usage, surgical interventions, and prolonged hospital stays appears fundamental.

Remarkably, our data correlated previous surgery and broad-spectrum antibiotic usage to systemic infection development. Furthermore, an appropriate colonisation index calculation may help clinicians to better classify high-risk patients for candidiasis [11,12]. Despite the absence of a clear statistical significance, our data demonstrated how the colonising *Candida* species is also the one with the higher probability of disseminating in proven candidiasis episodes. Despite careful colonisation and risk factor monitoring, invasive candidiasis remains a probable perspective in ICU patients. Colonisation indices support endogenous candidiasis prevention, but several exogenous sources (such as central venous catheters, parenteral nutrition, and hand transmission) persist within intensive care settings [14]. Therefore, a targeted clinical algorithm and a rapid diagnostic workflow seem essential. Molecular technologies decisively impact laboratory routine processes, which would require prolonged intervals if only based on gold-standard methods (microscopic and culture-based assays). According to our results, Real-Time PCR techniques allow for species identification without incubation for a prolonged time, expressing a significant precocity when compared to blood culture workflows and results.

The study aimed to evaluate the molecular assays as a possible integration for the current diagnostic workflow in suspected invasive candidiasis. Thus, we decided to compare the PCR results with the BDG results and blood culture data. The comparison revealed that positive blood culture results had an optimal agreement with the PCR reports in the presence of a positive BDG value.

Remarkably, the data show that all of the proven candidiasis cases had a molecular positive report, highlighting a 100% agreement between the culture-isolated *Candida* species and the PCR-detected species. Previous studies documented PCR’s high sensitivity and accuracy. These data enhanced the importance of applying a sensitive molecular assay to the diagnostic workflow [6,15]. Moreover, scientific evidence previously demonstrated some issues in BDG single usage, suggesting its combination with Real-Time PCR protocols in the IC diagnosis [16]. However, we selected only positive BDG samples for a PCR assay, demonstrating how this serological result may lack specificity. Specifically, several BDG-positive samples reported a negative PCR result, along with the absence of any culture isolation. Despite the low specificity rate, the BDG has a high sensitivity and a concentration distribution strictly correlated to the clinical condition. Notably, the BDG concentration reached high levels in candidiasis patients, demonstrating low rates among non-candidiasis patients. The BDG result represents the unique serological marker that is allowed to contribute to the invasive candidiasis diagnostic definition [5]. Our data confirm how duplicate BDG-positive values may significantly correlate with a probable/proven IC diagnosis due to the higher specificity rates than a single result. Indeed, the single result often suffers from false positivity rates (81.4%) that are higher than the PCR assays (5.7%). All of the probable-candidiasis patients reported at least two consecutive BDG-positive values, while only two proven candidiasis patients revealed a single BDG-positive value together with a positive PCR result.

These observations enhance the BDG and OLM CandID Real-Time PCR application on the same serum samples as an optimal strategy for the IC diagnosis, avoiding a second BDG detection for the real positivity confirmation. The OLM CandID Real-Time PCR method reveals sensitivity and specificity rates higher than the BDG results on the serum samples, even in duplicate BDG execution. Three patients with probable candidiasis revealed a molecular negative result, slightly impacting the overall statistical rates. It is fundamental to specify that our retrospective analysis included serum samples from previous refrigerated storage, which could have affected the global results, especially for negative results in high-risk candidiasis patients. Despite the interesting collected data, our study suffered from a significant limitation. Specifically, we analysed a limited number of included patients and found few proven candidiasis cases.

Invasive candidiasis may correlate with negative blood cultures, especially in the case of deep-sited infections [17,18,19,20,21,22]. Previously published data highlight how molecular methodologies can increase the diagnostic performance in terms of the sensitivity [23,24,25]. Furthermore, innovative technologies such as magnetic resonance have already integrated invasive candidiasis diagnosis in recent years [25,26]. The performance of OLM CandID Real-Time PCR is similar to that of other molecular methods. Despite this similarity, the tested kit expresses a wider identifiable species range [23].

Despite the interesting principles of magnetic resonance, its application requires 3–5 h to gather results and a specific biological sample (whole blood). The negative predictive value, sensitivity, and specificity may considerably vary when using this method. Consequently, a fungal species-identifying molecular assay, such as the OLM CandID Real-Time PCR, in the same workflow of a BDG result could optimise the diagnostic process.

This method may be applied on the same BDG serum sample, requiring approximately one hour to furnish a molecular result. Furthermore, we decided to test this kit due to its extended identification panel, including *C. dubliniensis*-specific probes. However, the molecular results confirmed the expected *Candida* spp. diffusion within our epidemiological area [27]. The study indeed reported *C. albicans* as the main invasive candidiasis aetiological agent.

This optimisation may improve the therapeutical management and exogenous/endogenous infection source detection. Despite its poor statistical correlation with the candidiasis condition, the colonisation index (values higher than 0.5) demonstrated how the colonising species are the same as that reported in the PCR results among the 12 probable/proven-candidiasis patients. These considerations allow for the hypothesis about an endogenous infection source. A single *C. albicans*-colonised patient reported a *C. parapsilosis* PCR result, leading to the possibility of an exogenous infection source. On these premises, the final purpose of our experimental protocol was to propose the importance of an appropriate integration between all of the available diagnostic tools. This integration may overcome specificity or sensitivity limitations, guaranteeing early responses and superior therapeutical supervision. According to serum minimal invasive collection, the contextual use of BDG results and a PCR assay could optimise ICU management and monitoring in the case of strong invasive candidiasis suspicion due to clinical signs or significant risk factors.

## 5. Conclusions

Despite its non-inclusion in the official diagnostic algorithms, *Candida* PCR assays may represent useful diagnostic assistance tools when applied together with serological markers and culture-based assays. This consideration emerges from the high sensitivity rate of molecular methodologies combined with the significant negative predictive value in excluding invasive candidiasis in high-risk patients.

## Figures and Tables

**Figure 1 microorganisms-13-00674-f001:**
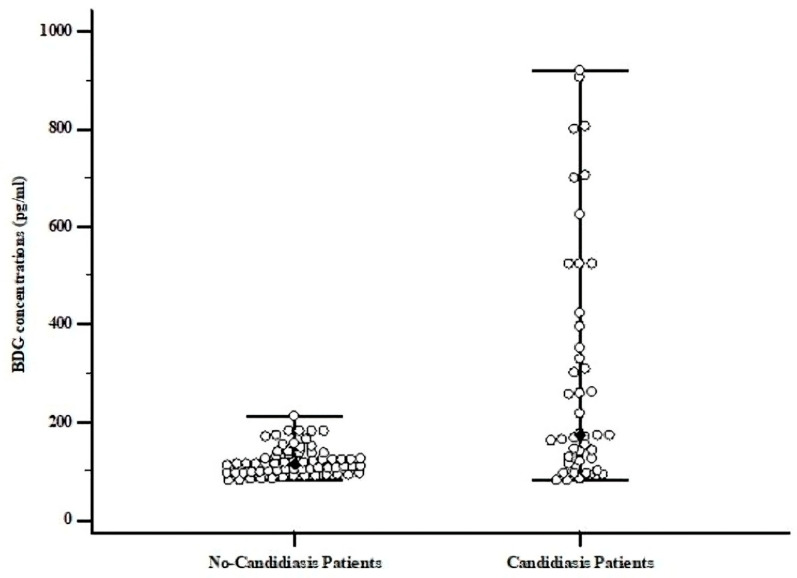
BDG distribution among no-candidiasis and candidiasis patients.

**Table 1 microorganisms-13-00674-t001:** Characteristics and risk factors of patients included in the study.

Patients and Risk Factors	Total (n = 60)	Proven IC (n = 7)	Probable IC (n = 10)	No IC (n = 43)	*p* ^a^
Male sex (%)	45 (75)	3 (42.8)	7 (70)	35 (81.4)	0.071
Age, median years (range)	71 (6 to 87)	71 (26 to 79)	70.5 (17 to 80)	71 (9 to 180)	0.669
ICU stay, median days (range)	34.5 (9 to 180)	40 (20 to 124)	44 (19 to 150)	30 (9 to 180)	0.135
PMV ^d^ (no., %)	9 (15.0)	2 (14.3)	2 (20)	6 (13.9)	0.703
Broad-spectrum antibiotics (no., %)	41 (68.3)	7 (100)	10 (100)	24 (55.8)	0.001
Any surgery under general anaesthesia (no., %)	13 (21.6)	4 (57.1)	3 (30)	6 (13.9)	0.022
Steroids (no., %)	33 (55)	2 (28.6)	6 (60)	25 (58.1)	0.440
Diabetes (no., %)	6 (10.0)	1 (14.3)	1 (10.0)	4 (9.3)	1.000
**No. of patients (%) with a:**					
Positive BG	52 (86.7)	7 (100)	10 (100)	35 (81.4)	0.091
Duplicate positive BG	25 (41.7)	5 (71.4)	10 (100)	10 (23.2)	<0.0001
Positive PCR ^b^	24 (46.1)	7 (100)	7 (70)	2 (5.7) ^c^	<0.0001
Colonisation index ≥ 0.5	27 (45)	4 (57.1)	5 (50)	18 (41.9)	0.440
Antifungal treatment	32 (53.3)	7 (100)	10 (100)	15 (34.9)	<0.0001

^a^ The *p* values were calculated by summing the proven and probable IC globally considered as candidiasis cases. ^b^ Positive PCR cases were calculated among the patients (52) with at least a positive BDG result. ^c^ Positive PCR cases were calculated among the non-candidiasis patients (35) with at least a positive BDG result. ^d^ Prolonged mechanical ventilation.

**Table 2 microorganisms-13-00674-t002:** Clinical and microbiological details about patients with proven/probable candidiasis.

No.	Underlying Conditions	Blood Culture	BDG ^a^	PCR ^b^	CI ^c^	*Candida* sp. Colonisation	LOS ^d^ ICU (Days)
3	Septic shock	Negative	154	*C. albicans*	>0.5	*C. albicans*	150
9	Respiratory failure	Negative	424	*C. albicans*	<0.5	*C. albicans*	42
11	Septic shock	Negative	350	*C. albicans*	>0.5	*C. albicans*	60
20	Septic shock	*C. parapsilosis*	804	*C. parapsilosis*	>0.5	*C. parapsilosis*	124
25	Septic shock	*C. albicans*	443	*C. albicans*	>0.5	*C. albicans*	22
27	Septic shock	Negative	523	Negative	>0.5	*C. glabrata*	36
31	Respiratory failure	Negative	262	*C. albicans*	>0.5	*C. albicans*	19
33	Kidney cancer, septic shock	*C. tropicalis*	920	*C. tropicalis*	>0.5	*C. tropicalis*	65
34	Chronic renal failure	Negative	396	Negative	-	-	46
37	Respiratory failure	Negative	357	*C. parapsilosis*	<0.5	*C. albicans*	55
47	Trauma	*C. parapsilosis*	704	*C. parapsilosis*	>0.5	*C. parapsilosis*	40
49	Renal transplant	*C. tropicalis*	176	*C. tropicalis*	<0.5	*C. tropicalis*	35
50	Haematological malignancy	Negative	523	*C. albicans*	>0.5	*C. albicans*	96
53	Respiratory failure	Negative	300	Negative	<0.5	*C. glabrata*	25
54	Respiratory failure	Negative	310	*C. albicans*	-	-	33
59	Intestinal occlusion	*C. albicans*	144	*C. albicans*	<0.5	*C. albicans*	20
60	Gastrointestinal perforation	*C. albicans*	259	*C. albicans*	<0.5	*C. albicans*	44

^a^ The highest BDG ((1→3)-β-D-glucan) value observed during the evaluation period; ^b^ Polymerase chain reaction by the OLM CandID Real-Time PCR assay; ^c^ Colonisation index; ^d^ Length of stay.

**Table 3 microorganisms-13-00674-t003:** Summary of the positive PCR assay cases, along with the blood culture results and the relative BDG values.

Techniques	Positive PCR Assay						
	*C. albicans*	*C. glabrata*	*C. parapsilosis*	*C. tropicalis*	*C. krusei*	Negative PCR	Total
**Blood Culture**							
*C. albicans*	3 (5.8%)	0	0	0	0	0	3 (5.8%)
*C. glabrata*	0	0	0	0	0	0	
*C. parapsilosis*	0	0	2 (3.8%)	0	0	0	2 (3.8%)
*C. tropicalis*	0	0	0	2 (3.8%)	0	0	2 (3.8%)
*C. krusei*	0	0	0	0	0	0	0
Negative	6 (11.5%)	0	1 (1.9%)	0	2 (3.8%)	36 (69.2%)	45 (86.5%)
**BG**							
Positive	9 (17.3%)	0	3 (5.8%)	2 (3.8%)	2 (3.8%)	36 (69.2%)	52 (100%)
Negative	0	0	0	0	0	0	0
**Total**	**9 (17.3)**	**0**	**3 (5.8%)**	**2 (3.8%)**	**2 (3.8%)**	**36 (69.2%)**	**52 (100%)**

## Data Availability

The present manuscript includes all of the data collected during the study. The study was conducted according to the guidelines of the Declaration of Helsinki and the best clinical practice (D.M. 15/07/1997). The present study does not directly involve patient management or drug administration. The studies were conducted in accordance with the local legislation and institutional requirements.

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
