# Peer review of "The Role of the OLM CandID Real-Time PCR in the Invasive Candidiasis Diagnostic Surveillance in Intensive Care Unit Patients"

_microorganisms, 2025, doi:10.3390/microorganisms13030674_

Round 1
Reviewer 1 Report (New Reviewer)
Comments and Suggestions for Authors
Introduction
L26 – Which mucosal anatomical sites can be affected? Please describe.
I suggest presenting data on the prevalence, mortality, and costs associated with managing Candida infections.
Why are you testing a new strategy (OLM CandID Real-61 Time PCR) when magnetic resonance and Real-Time PCR already demonstrated good sensitivity for Candida detection?
What characteristics of OLM CandID Real-61 Time PCR differentiate it from other qPCR methods? Please provide detailed information.
Materials and Methods
What are the exclusion criteria?
The nomenclature of several Candida species has been updated, including Candida glabrata and Candida krusei. Please update this information.
L126 – What are the confidence intervals for the other assays?
I suggest dividing the Materials and Methods section into subsections, such as ethical aspects, inclusion and exclusion criteria, sample collection, etc.
What is CandID? Please provide a detailed description of the method. Reference if it has already been used/published.
Results
Table 1 – Why was the parameter “Male” sex used? Is there is any relationship between male sex and the occurrence of candidiasis?
The caption of Figure 1 is described as Graph 1. Please correct this inconsistency.
Was the BDG distribution between patients with and without candidiasis statistically different?
What is the sensitivity, specificity, PPV and PVN of CandID? Are there statistical differences between CandID, qPCR, and BGR?
The authors must clarify where traditional qPCR or CandID was used. In all instances where PCR is mentioned, should it be understood as referring to CandID? This information is unclear.
The data should be explored in more depth. What is the prevalence of the Candida species detected? At what point during the sample collection were samples positive?
Discussion
What is the sensitivity of OLM CandID Real-61 Time PCR for species related to Candida genus (Candidozyma auris, Meyerozyma guilliermondii, and others)? Since only species of the genus Candida were detected (and none from related genera), could this be considered a limitation of the method?
The authors propose to study “the role of the OLM CandID Real-61 Time PCR in the early and accurate candidemia diagnosis 62 among ICU patients”. How can they state that the strategy was adequate for detecting early infection when the kinetics of sample collection after infection were not presented?
Discuss the prevalence of Candida species found in this study. Is it expected?
Please italicize scientific names throughout the manuscript.
The authors proposed a new method. What are the advantages and disadvantages of this method compared to other methods, particularly qPCR? This information needs to be fully discussed.
Compare the findings with other studies that analyzed new experimental diagnostic approaches for identifying fungal agents.
Comments on the Quality of English Language
The English could be improved in some points of the manuscript.
Author Response
Introduction
Comment: L26 – Which mucosal anatomical sites can be affected? Please describe.
Answer: The sentence has been revised.
Comment: I suggest presenting data on the prevalence, mortality, and costs associated with managing Candida infections.
Answer: We added some sentences within the introduction (Please, see reference n. 2)
Comment: Why are you testing a new strategy (OLM CandID Real-Time PCR) when magnetic resonance and Real-Time PCR already demonstrated good sensitivity for Candida detection?
Answer: We added the following paragraph within the discussion section: “Despite magnetic resonance interesting principles, its application requires 3-5 hours to gather results and a specific biological sample (whole blood). Consequently, a fungal species identifying molecular assay such as the OLM CandID Real-Time PCR in the same workflow of a BDG result could optimize the diagnostic process. This method may be applied on the same BDG serum sample, requiring approximately one hour to furnish a molecular result. Furthermore, we decided to test this kit due to its extended identification panel, including C. dubliniensis specific probes”. These sentences aim to justify the importance to investigate molecular methods other than magnetic resonance.
Comment: What characteristics of OLM CandID Real-Time PCR differentiate it from other qPCR methods? Please provide detailed information.
Answer: We discussed some differences between several molecular methods within the discussion section. Unfortunately, we did not make any comments about quantitative PCR due to the commercial absence of quantitative Candida spp. DNA detection kits.
Materials and Methods
Comment: What are the exclusion criteria?
Answer: We added some sentences to clarify the exclusion criteria within the materials and methods section: “Patients with ICU stays equal or shorter than 4 days were excluded from the study. Additionally, patients with prolonged ICU stay (>4 days) in the absence of at least one of the above-mentioned risk factors were not included”.
Comment: The nomenclature of several Candida species has been updated, including Candida glabrata and Candida krusei. Please update this information.
Answer: Thank you for the observation. We added the requested information.
Comment: L126 – What are the confidence intervals for the other assays?
Answer: The sentence about confidence intervals was a typo. It has been removed.
Comment: I suggest dividing the Materials and Methods section into subsections, such as ethical aspects, inclusion and exclusion criteria, sample collection, etc.
Answer: Thank you for the fundamental suggestion. We applied the modifications.
Comment: What is CandID? Please provide a detailed description of the method. Reference if it has already been used/published.
Answer: “CandID” and “OLM CandID Real-Time PCR” are the same thing. We specified only the expression “OLM CandID Real-Time PCR” to avoid misunderstandings along the manuscript.
Results
Comment: Table 1 – Why was the parameter “Male” sex used? Is there is any relationship between male sex and the occurrence of candidiasis?
Answer: Yes, literature data demonstrated that male sex may be considered as an invasive risk factor for invasive candidiasis. Please, see the following reference: “Egger M, Hoenigl M, Thompson GR 3rd, Carvalho A, Jenks JD. Let's talk about sex characteristics-As a risk factor for invasive fungal diseases. Mycoses. 2022 Jun;65(6):599-612. doi: 10.1111/myc.13449. Epub 2022 May 25. PMID: 35484713”. This reference has been added to the manuscript (reference number 13).
Comment: The caption of Figure 1 is described as Graph 1. Please correct this inconsistency.
Answer: We are sorry for the typo. We corrected it.
Comment: Was the BDG distribution between patients with and without candidiasis statistically different?
Answer: Yes, the BDG distribution illustrated higher BDG concentration in candidiasis patients than in no-candidiasis episodes. We specified this details within the results.
Comment: What is the sensitivity, specificity, PPV and PVN of CandID? Are there statistical differences between CandID, qPCR, and BGR?
Answer: Please, note that CandID is the OLM CandID Real-Time PCR; thus, we don’t need to indicate any differences between CandID and PCR assays. What is the meaning of the acronym BGR? We hypothesized you meant “BDG”. As a conclusion, all the details about sensitivity, specificity, PPV, and NPV have been already indicated for both molecular and serological results within the manuscript.
Comment: The authors must clarify where traditional qPCR or CandID was used. In all instances where PCR is mentioned, should it be understood as referring to CandID? This information is unclear.
Answer: We specified this details answering to previous comments.
Comment: The data should be explored in more depth. What is the prevalence of the Candida species detected? At what point during the sample collection were samples positive?
Answer: We added some details about the Candida species prevalence among candidiasis cases. On the other hand, we performed a retrospective analysis, so we can only document that serum samples were collected at the same time of the blood culture among candidiasis cases. We added this detail within the materials and method (subsection 2.2).
Discussion
Comment: What is the sensitivity of OLM CandID Real-Time PCR for species related to Candida genus and others)? Since only species of the genus Candida were detected (and none from related genera), could this be considered a limitation of the method?
Answer: Candida auris (Candidozyma auris), Candida guilliermondii (Meyerozyma guilliermondii), and other Candida species diverse than Candida albicans, Candida glabrata, Candida parapsilosis, Candida krusei, Candida dubliniensis, and Candida tropicalis are not included within the OLM CandID Real-Time PCR identification spectrum. We specified the identifiable species within the Materials and methods section.
Comment: The authors propose to study “the role of the OLM CandID Real-61 Time PCR in the early and accurate candidemia diagnosis 62 among ICU patients”. How can they state that the strategy was adequate for detecting early infection when the kinetics of sample collection after infection were not presented?
Answer: We could not provide information about the kinetics because the study was planned as a retrospective analysis only on biological samples. Consequently, we hypothesized the use of the molecular assay as a promising strategy, aiming to enforce the hypothesis through further clinical studies directly involving patients’, clinical prospective evaluation, and therapeutical choices. Please, see the use of expressions such as “may be”, “could optimize”, and “the final purpose was to propose” as a confirmation of the cautious conclusions we gathered from this preliminary study.
Comment: Discuss the prevalence of Candida species found in this study. Is it expected?
Answer: Yes, it was expected to find C. albicans as the main aetiological agent. We provided some sentences to highlight this consideration within the discussion.
Comment: Please italicize scientific names throughout the manuscript.
Answer: All the scientific names have been revised.
Comment: The authors proposed a new method. What are the advantages and disadvantages of this method compared to other methods, particularly qPCR? This information needs to be fully discussed.
Answer: We revised some sentences within the discussion comparing the tested PCR to other technologies, such as the T2 magnetic resonance. However, we did not discuss anything about qPCR due to the absence of diagnostic validated quantitative PCR tests for Candida spp. DNA detection.
Comment: Compare the findings with other studies that analyzed new experimental diagnostic approaches for identifying fungal agents.
Answer: We added some sentences within the discussion to integrate the comparison between the OLM CandID Real-Time PCR and other existing molecular methods in Candida spp. DNA detection.

Reviewer 2 Report (New Reviewer)
Comments and Suggestions for Authors
60 suitable patients [patients with prolonged ICU stay (>4 days) 69 and the presence of one or more risk factors such as broad-spectrum antibiotic therapy, 70 haematological diseases, prolonged corticosteroid usage, and Candida colonization index 71 (CI) value ≥ 0.5]
10 probable 15 and 7 proven candidiasis cases – the abstract needs to state that this is a retrospective study. Also, line 12 “..the early and accurate candidaemia diagnosis..” suggest “..the early and rapid candidaemia diagnosis..” as you have not yet demonstrated accuracy.
Please specify if you use any infection prevention interventions in your ICU. Specifically, due you use topical antimicrobial interventions? Even though these are not broad spectrum, they may increase Candidemia and colonizaiton [Hurley J. Structural Equation Modelling as a Proof-of-Concept Tool for Mediation Mechanisms Between Topical Antibiotic Prophylaxis and Six Types of Blood Stream Infection Among ICU Patients. Antibiotics. 2024 Nov 18;13(11):1096
Hurley JC. Impact of selective digestive decontamination on respiratory tract Candida among patients with suspected ventilator-associated pneumonia. A meta-analysis. European Journal of Clinical Microbiology & Infectious Diseases. 2016 Jul;35(7):1121-35.].
Why is the LOS so long [>30 days].
How many were receiving mechanical ventilation?
How many were receiving empiric anti-fungal therapy?
How many had diabetes?
Author Response
- Comment: 60 suitable patients [patients with prolonged ICU stay (>4 days) 69 and the presence of one or more risk factors such as broad-spectrum antibiotic therapy, 70 haematological diseases, prolonged corticosteroid usage, and Candida colonization index 71 (CI) value ≥ 0.5]. 10 probable 15 and 7 proven candidiasis cases – the abstract needs to state that this is a retrospective study. Also, line 12 “…The early and accurate candidaemia diagnosis...” suggest “. “The early and rapid candidaemia diagnosis...” as you have not yet demonstrated accuracy.
Answer: We modified the above-mentioned details. Thank you for the suggestions. - Comment: Please specify if you use any infection prevention interventions in your ICU. Specifically, due you use topical antimicrobial interventions? Even though these are not broad spectrum, they may increase Candidemia and colonization. (Hurley J. Structural Equation Modelling as a Proof-of-Concept Tool for Mediation Mechanisms Between Topical Antibiotic Prophylaxis and Six Types of Blood Stream Infection Among ICU Patients. Antibiotics. 2024 Nov 18;13(11):1096); (Hurley JC. Impact of selective digestive decontamination on respiratory tract Candida among patients with suspected ventilator-associated pneumonia. A meta-analysis. European Journal of Clinical Microbiology & Infectious Diseases. 2016 Jul;35(7):1121-35).
Answer: We already specified the requested information. Table 1 includes broad spectrum antibiotics usage and its statistical significance correlating this risk-factor to invasive candidiasis. - Comment: Why is the LOS so long [>30 days].
Answer: We included intensive care unit patients, whose LOS is normally prolonged due to chronic disease, frequent unconsciousness, and respiratory failure (which require mechanical ventilation). - Comment: How many were receiving mechanical ventilation?
Answer: We added a statistic analysis about prolonged mechanical ventilation within Table 1. - Comment: How many were receiving empiric anti-fungal therapy?
Answer: The data collection documented the empiric antifungal treatment for twenty-two patients, among which seven patients were included within the proven/probable candidiasis cases. - Comment: How many had diabetes?
Answer: We added a statistic analysis about diabetes within Table 1.

Round 2
Reviewer 1 Report (New Reviewer)
Comments and Suggestions for Authors
NA
This manuscript is a resubmission of an earlier submission. The following is a list of the peer review reports and author responses from that submission.
Round 1
Reviewer 1 Report
Comments and Suggestions for Authors
The paper needs significant revision especially because of English language and of the discussion which should be more clear and concentrated upon the subject of the study.

English needs a lot of work in order the reader understand what the authors mean
Author Response
Comment: The paper needs significant revision especially because of English language and of the discussion which should be clearer and more concentrated upon the subject of the study. English needs a lot of work in order the reader understand what the authors mean.
Answer: Thank you for the suggestion. The English language has been revised. Please, find all the changes highlighted through the tracking mode into the manuscript file.
Reviewer 2 Report
Comments and Suggestions for Authors
The authors have examined the diagnostic performance of the OLM (now Immy) CandID CE-marked PCR kit for diagnosing candidaemia and possibly invasive candidiasis. Overall the kit performed very well, in people with at least 1 positive BDG assay.
Comments.
1. There were 2 PCR positive cases with also a positive BDG amongst the non-IC cases. These cases have determined the specificity of the test, but these may be IC cases, which is difficult to diagnose. The authors need to provide more detail about these cases (including their colonisation indices, before concluding that they are ‘false positive’ cases. Blood culture is about 40% sensitive for IC.
2. The abstract should include the key information about 100% species concordance with positive blood cultures, and identified the causative species in 7 additional cases.
3. The CandID assay takes about 45 mins from extraction. A comment in discussion on the time frames for BDG and PCR vs blood culture would be helpful, with their experience in a diagnostic lab. If they have turnaround time data, it should be inserted into the results.
4. Final sentence of the conclusion is too cautious. The performance was better than blood culture. The authors can be bold.
5. L38 Blood culture for IC is poor (see above) and cannot be considered a ‘gold’ standard – perhaps silver or bronze standard. See Nguyen Figure 1 (doi: 10.1093/cid/cis200) and many other papers.
Minor comments
They do not explain how CandID PLUS was used.
No cases of C. auris – a limitation in some places.
Comments on the Quality of English LanguageSome English phrasing not ideal. Examples include L15 (di-agnostic), L41 (gathers), L52 (‘title’), L53 ‘The fungal DNA detection’, etc. It would be helpful for a native English writer to provide some input.
Author Response
- Comment: There were 2 PCR positive cases with also a positive BDG amongst the non-IC cases. These cases have determined the specificity of the test, but these may be IC cases, which is difficult to diagnose. The authors need to provide more detail about these cases (including their colonisation indices, before concluding that they are ‘false positive’ cases. Blood culture is about 40% sensitive for IC.
Answer: We could not classify the above-mentioned patients as candidiasis episodes according to the EORTC criteria. Specifically, those criteria state that proven candidiasis has positive Candida blood culture result. Otherwise, probable candidiasis was defined by mycological evidence (at least two consecutive positive BDG detections or positive magnetic resonance Candida results), host factors (haematological diseases, prolonged corticosteroids usage, Candida colonization, documented organ transplants), and clinical features (organs’ abscesses or lesions). Patients who did not satisfy these criteria were not included among the proven or probable candidiasis cases. However, the expression “false positive” may be incoherent and we decided to remove it according to your suggestion. - Comment: The abstract should include the key information about 100% species concordance with positive blood cultures, and identified the causative species in 7 additional cases.
Answer: thank you for the suggestion. We added a sentence about this information within the abstract. - Comment: The CandID assay takes about 45 mins from extraction. A comment in discussion on the time frames for BDG and PCR vs blood culture would be helpful, with their experience in a diagnostic lab. If they have turnaround time data, it should be inserted into the results.
Answer: Few sentences have been added to comment these aspects within the discussion. - Comment: Final sentence of the conclusion is too cautious. The performance was better than blood culture. The authors can be bold.
Answer: Thank you for the comment. The conclusion section has been revised. - Comment: Blood culture for IC is poor (see above) and cannot be considered a ‘gold’ standard – perhaps silver or bronze standard. See Nguyen Figure 1 (doi: 10.1093/cid/cis200) and many other papers.
Answer: We revised the references about this topic, adding the above-mentioned paper to enlarge the knowledge. Furthermore, we decided to remove the expression “gold-standard” according to your suggestion.
Please, find all the requested changes highlighted in yellow within the manuscript.
Minor comments
- Comment: They do not explain how CandID PLUS was used.
Answer: We apologize for the missing information. It has been added within the Materials and Methods chapter. - Comment: No cases of auris – a limitation in some places.
Answer: We agree with you about this comment. We added some words to comment this limitation within the discussion chapter.
Please, find all the requested changes highlighted in yellow within the manuscript.
Comments on the English language
- Comment: Some English phrasing not ideal. Examples include L15 (di-agnostic), L41 (gathers), L52 (‘title’), L53 ‘The fungal DNA detection’, etc. It would be helpful for a native English writer to provide some input.
Answer: Thank you for the suggestion. The English language has been revised. Please, find all the changes highlighted through the tracking mode into the manuscript file.
Round 2
Reviewer 1 Report
Comments and Suggestions for Authors
I insist on my previous comments
Comments on the Quality of English LanguageNeeds correction
Author Response
We considered all the comments, revising all the introduction and discussion paragraphs. Furthermore, we changed some sentences within the results. The definitive purpose was to address the requests about clearer paragraphs and better English language. Please, find all the requested modifications highlighted in yellow.